# The Epidemiology of Multidrug-Resistant Sepsis among Chronic Hemodialysis Patients

**DOI:** 10.3390/antibiotics11091255

**Published:** 2022-09-15

**Authors:** Shani Zilberman-Itskovich, Yazid Elukbi, Roni Weinberg Sibony, Michael Shapiro, Dana Zelnik Yovel, Ariela Strulovici, Amin Khatib, Dror Marchaim

**Affiliations:** 1Tel-Aviv Medical Center (Sourasky), Tel-Aviv 6423906, Israel; 2Sackler School of Medicine, Tel-Aviv University, Tel-Aviv 6997801, Israel; 3Shamir (Assaf Harofeh) Medical Center, Zerifin 70300, Israel; 4Ben Gurion University, Beer-Sheva P.O. Box 653, Israel

**Keywords:** hemodialysis, sepsis, antimicrobial resistance, MDRO, dialysis

## Abstract

Sepsis is one of the leading causes of hospitalization and death among hemodialysis patients. Infections due to multidrug-resistant organisms (MDROs) are common among these patients, but empiric broad-spectrum coverage for every septic patient is associated with unfavorable outcomes. A retrospective case–control study was conducted at Shamir Medical Center, Israel (July 2016–April 2020), to determine predictors of MDRO infections among septic (per SEPSIS-3) ambulatory adult hemodialysis patients with permanent dialysis access (i.e., fistula, graft, or tunneled Perm-A-Cath). MDROs were determined according to established definitions. Least Absolute Shrinkage and Selection Operator (LASSO) regression was used to construct a prediction score and determine its performance. Of 509 patients, 225 (44%) had microbiologically confirmed infection, and 79 patients (35% of 225) had MDROs. The eventual independent predictors of MDRO infections were Perm-A-Cath access (vs. fistula or graft, aOR = 3, CI-95% = 2.1–4.2) and recent hospitalization in the previous three months (aOR = 2.3, CI-95% = 1.6–3.3). The score to predict MDRO sepsis with the highest performances contained seven parameters and displayed an area under the receiver operating characteristic curve (ROC AUC) of 0.74. This study could aid in defining a group of hemodialysis patients for which empiric broad-spectrum agents could be safely avoided.

## 1. Introduction

Sepsis is one of the leading causes of hospitalization and death among chronic hemodialysis patients [1]. The high incidence of infections among this population is related to immunosuppressive and immunomodulatory states and conditions, constant access to a large blood vessel, and extensive and constant exposures to healthcare settings, environments, and procedures [1,2].

Infections due to multidrug-resistant organisms (MDROs) were defined by the World Health Organization (WHO) as one of the current greatest challenges in Medicine [3]. MDRO infections are relatively prevalent among dialysis patients [4], since these patients are subjected to high colonization pressure (facilitating “patient-to-patient transmission” of MDROs) and high selective pressure (facilitating “emergence of resistance” among the patient’s own non-MDRO susceptible strains) [4]. Delay in administration of appropriate antimicrobial therapy (DAAT) is common in MDRO infections [5] and is the strongest independent modifiable predictor of mortality in septic shock [5]. Therefore, it is important to avoid DAAT among certain hemodialysis septic patients suspected of having MDRO infection while avoiding usage of overly broad-spectrum coverage among hemodialysis patients at lower risk for MDROs [4]. Our study aim was to explore the epidemiology of MDRO sepsis among chronic ambulatory hemodialysis patients.

## 2. Results

There were 509 chronic ambulatory hemodialysis patients with sepsis upon admission to Shamir Medical Center (SMC) during the study period. Most patients were elderly (i.e., 71%, mean age 71 ± 13 years), and 67% were male. One-hundred and sixty-six patients (33%) were already flagged as known MDRO carriers in the previous 24 months. The median dialysis vintage (i.e., duration) was 2 years (range 0–31 years). With regard to the infectious syndromes of acute sepsis, 169 patients (33%) had respiratory infections, and 163 (32%) had primary bloodstream infections (i.e., BSI). The median length of stay in the hospital was eight days (IQR 4–13 days). Sixty-four patients (13%) died in the hospital, and 125 (25%) died within 90 days.

The microbiological diagnosis was confirmed for 225 (44%) patients (Table 1). *Staphylococcus aureus* was the commonest pathogen (17.8%), followed by *Escherichia coli* (16%) and *Pseudomonas aeruginosa* (9.8%). MDROs were documented in 79 patients (35% of microbiologically confirmed cases), and the commonest MDROs were third-generation-cephalosporin-non-susceptible Enterobacterales (i.e., extended-spectrum beta-lactamase (ESBL) and/or *bla*_AmpC_-hyperproducing strains, 16%), *Pseudomonas aeruginosa* (10%), and methicillin-resistant *S. aureus* (i.e., MRSA, 5%).

Empiric usage of inappropriate agents (i.e., either an overly broad or narrow spectrum, or completely “wrong” therapy) was documented among 105 patients, i.e., 47% of the patients with microbiologically confirmed infection.

Next, we analyzed the predictors of MDRO infections. In univariable analyses (Table 2), there were multiple parameters associated with MDRO sepsis. Although demographic features (age, sex, and place of residence) did not differ between groups, there were multiple other significant statistical associations and potential predictors of MDRO sepsis, i.e., Perm-A-Cath access (vs. fistula or graft), certain background comorbidities (e.g., peripheral arterial disease and chronic skin ulcers), recent healthcare-associated exposures (i.e., hospitalizations, procedures, devices, and antibiotics), and certain acute illness indices (e.g., tachycardia at presentation, fatal McCabe score, low serum albumin, and admission to intensive care unit). The portion of patients with MDRO sepsis who were treated with appropriate antibiotics (per in vitro report) in the first 48 h (73%) was significantly lower in comparison to patients with non-MDRO sepsis (86%; OR = 0.4, *p* = 0.02) and experienced significantly worse outcomes (Table 2). 

In a multivariable model, the only independent significant predictors of MDRO sepsis remained Perm-A-Cath access (vs. fistula or graft, aOR = 3, CI-95% = 2.1–4.2) and recent hospitalization in the previous three months (aOR = 2.3, CI-95% = 1.6–3.3). The negative predictive value (NPV), if neither one of these parameters was present, was 91%.

The score to predict MDRO sepsis with the highest performance contained seven parameters: (1) tachycardia (2 points), (2) antibiotic use in the previous 3 months (2 points), (3) known MDRO carriage in the past two years (3 points), (4) chronic skin ulcers (3 points), (5) admission to ICU (4 points), (6) skilled nursing care at home (4 points), and (7) Perm-A-Cath access (4 points). A cutoff of 6 points yielded 75% sensitivity, 60% specificity, 31% positive predictive value (PPV), and 91% NPV, with a ROC AUC of 0.74 (Figure 1).

This plot demonstrates the receiver operating characteristic curve analysis (ROC) for the prediction performance of the score to predict multidrug-resistant organism infection among chronic hemodialysis patients.

There were certain outcomes among survivors that were significantly worse among MDRO patients, including elongated length of stay (LOS; 10 vs. 7 days, *p* = 0.001) and functional status deterioration following the index event (23% vs. 11%, OR = 2.5, *p* = 0.005). The in-hospital mortality rate was higher as well (though insignificantly) among patients with MDRO infections (19% vs. 11%, *p* = 0.06). In separate multivariable models, one for each outcome parameter, MDRO remained significantly associated with elongated LOS (i.e., LOS > 10 days) following the infection (aOR = 2, CI-95% = 1.1–3.6, *p* = 0.03), but not with functional status deterioration nor with in-hospital mortality.

## 3. Discussion

In this study, among a cohort of 509 patients, MDROs were documented among 79 patients, i.e., 35% of the 225 patients with microbiologically confirmed infection. Moreover, one of every three patients was already a known MDRO carrier upon admission. This indicates a large burden of MDROs, implying that MDROs are endemic among chronic ambulatory hemodialysis patients admitted with acute sepsis to SMC. This theoretically justifies the current practices of empiric administration of broad-spectrum agents for hemodialysis patients presenting with acute sepsis to an acute-care facility. Nonetheless, we were able to isolate a group of hemodialysis patients in whom overly broad-spectrum (and frequently toxic) agents could be safely avoided. Despite the endemicity of MDROs at SMC, 99 of 146 (68%) patients with microbiologically confirmed susceptible organism (non-MDRO) sepsis had empirically received an overly broad-spectrum agent (Table 2). The use of overly broad-spectrum antibiotics for every septic patient further increases the selective pressure among this population and increases the rates of adverse events, which are frequently elevated among patients with reduced kidney function who receive broad-spectrum non-beta-lactam regimens [15].

In a multivariable model, the only independent significant predictors of MDROs were Perm-A-Cath access and recent acute-care hospitalization (in the past 3 months). If neither one of these parameters was present, the negative predictive value (NPV) for having an MDRO infection was as high as 91%. Among this group of chronic ambulatory hemodialysis patients, empiric broad-spectrum coverage could be safely avoided among patients with non-life-threatening infections. Next, we developed a prediction score to further improve MDRO prediction performance, but while the NPV of the seven-component score was the same (i.e., 91%), the overall ROC AUC was only 0.74. Therefore, specifically at SMC, the absence of the two independent predictors of MDRO infection (i.e., no Perm-A-Cath access and no recent hospitalizations) should guide the empiric management to avoid overly broad-spectrum agents among certain patients, and this should not be based on the incorporation of a complicated score with difficult assimilation challenges among busy healthcare workers.

Our study has limitations associated with its retrospective chart-review design from a single center. The conclusions of this study could not be generalized to additional facilities and centers without conducting internal validations. However, as opposed to previous MDRO epidemiology analyses executed among ambulatory hemodialysis patients [16], this analysis included not only patients with BSI but all septic patients (per SEPSIS-III definition) [17] while also attributing offending pathogens to non-BSI infectious syndromes, determined by a single senior Infectious Disease specialist. This design better reflects the spectrum and complexities of decisions that practitioners face while attending a hemodialysis patient with acute sepsis.

To conclude, at SMC, MDRO infections are prevalent among chronic ambulatory hemodialysis patients, and it is a safe routine to empirically administer broad-spectrum agents to the majority of septic patients. However, we were able to identify a group of patients in whom overly broad-spectrum coverage (i.e., frequently more toxic, less bactericidal, and more expensive alternatives) could be safely avoided. This could lead to improved outcomes among these individuals and could result, over time, in beneficial institutional ecological impacts.

## 4. Materials and Methods

This was a retrospective cohort study at Shamir Medical Center (SMC, July 2016–April 2020), Israel. The ambulatory hemodialysis department at SMC hosts ~200 chronic hemodialysis inpatients and outpatients at any given time. The institutional ethics committee at SMC had approved the study prior to its initiation. Adult patients (>18 years) on chronic hemodialysis with acute sepsis [17] upon admission to SMC’s emergency room (ER) were enrolled. Patients on acute dialysis, patients with nosocomial sepsis, and patients with a non-tunneled temporary line (i.e., no arteriovenous fistula/graft or tunneled line (e.g., Perm-A-Cath)) were all excluded. Patients could be included more than once, but only if the septic episodes were separated by more than one month, and only if the patient presented with a different infectious syndrome. The clinical infectious syndrome was determined according to established guidelines [18]. We captured cultures that were obtained from sterile sites (e.g., blood) or cultures that were obtained from non-sterile sites but matched the patient’s infectious syndrome (e.g., respiratory cultures among patients with pneumonia and urine cultures among patients with urinary tract infections).

MDROs were defined in accordance with established definitions [19]: (1) methicillin-resistant *Staphylococcus aureus* (MRSA); (2) ampicillin-resistant enterococci; (3) penicillin- or ceftriaxone-non-susceptible *Streptococcus pneumoniae*; (4) *P. aeruginosa*; (5) *A. baumannii*; (6) Enterobacterales non-susceptible to one or more 3rd-generation cephalosporins (e.g., ceftriaxone or ceftazidime); and (7) fluoroquinolone-resistant *Campylobacter jejuni*. Extensively drug-resistant organisms (XDROs) included: (1) vancomycin-resistant enterococci (VRE); (2) heterogeneous vancomycin-intermediate *S. aureus* (hVISA) or *S. aureus* with MIC ≥ 2 to vancomycin; (3) carbapenem-resistant Enterobacterales (i.e., CRE, evidence of carbapenemase production and/or meropenem MIC ≥ 2); (4) carbapenem-resistant *A. baumannii* (CRAB); (5) carbapenem-resistant *P. aeruginosa* (CRPA); and (6) intrinsically carbapenem-non-susceptible Gram-negatives (e.g., *Stenotrophomonas maltophilia*). Microbiological processing was in accordance with Clinical and Laboratory Standards Institute (CLSI) criteria and definitions [20].

Predictors of MDRO sepsis were queried by logistic regression. We randomized the cohort into two groups: data from 80% of the patients were used in order to generate and create the prediction score, and data from 20% of the patients were used in order to validate the prediction score that was generated. A prediction score of MDRO sepsis was developed using Least Absolute Shrinkage and Selection Operator (LASSO) regression. In short, the L1-penalty was used to determine the model with the best performance while penalizing the sum of absolute values of the coefficients. The coefficients for the least useful variables for prediction were driven to 0 and were excluded from further analysis. The Python “scikit-learn” machine learning library was used in order to perform the LASSO regression. The remaining variables were used in the multivariable logistic regression. The variables that were significant were used in order to construct the final predictive score. The score was then evaluated using a validation cohort that was not used in the development of the score.

## Figures and Tables

**Figure 1 antibiotics-11-01255-f001:**
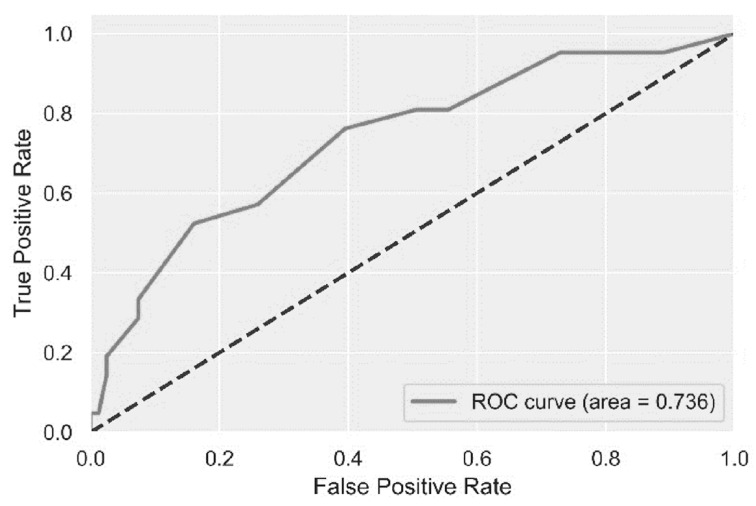
A prediction score for multidrug-resistant organism (MDRO) infection among chronic ambulatory adult hemodialysis patients (Shamir Medical Center, July 2016–April 2020).

**Table 1 antibiotics-11-01255-t001:** Chronic ambulatory hemodialysis patients with microbiologically confirmed sepsis upon admission to Shamir Medical Center (July 2016–April 2020).

Pathogen	Number (Valid Percent ^1^)	Infectious Clinical Syndrome
**Gram-Positives**		
*Staphyloccocus aureus*	40 (17.8%)	Primary BSI (28, 12%), SSTI (9, 4%), respiratory (2, 1%), and intra-abdominal infection (1, 0.5%).
Coagulase-negative Staphylococci ^2,3^	12 (5%)	Primary BSI (9, 4%) and other (3, 1%).
*Enterococcus faecalis*	11 (5%)	Primary BSI (6, 3%), other (2, 1%), UTI (2, 1%), and intra-abdominal infection (1, 0.5%).
*Enterococcus faecium*	1 (0.5%)	SSTI (1, 0.5%).
Other *Enterococcus* species	2 (1%)	SSTI (1, 0.5%). Intra-abdominal infection (1, 0.5%).
*Streptococcus pyogenes*	4 (2%)	SSTI (3, 1%) and respiratory (1, 0.5%).
*Streptococcus pneumoniae*	3 (1%)	Respiratory (2, 1%) and other (1, 0.5%).
Streptococci bovis group ^4^	6 (3%)	Primary BSI (4, 2%) and UTI (2, 1%).
*Streptococcus agalactiae*	3 (1%)	Primary BSI (2, 1%) and SSTI (1, 0.5%).
Streptococci viridans group ^5^	3 (1%)	Primary BSI (1, 0.5%) and other (2, 1%)
*Corynebacterium* spp. ^2^	2 (1%)	Primary BSI (2, 1%).
*Lactobacillus* spp. ^2^	1 (0.5%)	Primary BSI (1, 0.5%).
**Gram-Negatives**		
*Escherichia coli*	36 (16%)	UTI (12, 5%), SSTI (8, 4%), respiratory (8, 4%), intra-abdominal infection (7, 3%), and primary BSI (1, 0.5%).
*Pseudomonas aeruginosa*	22 (10%)	Primary BSI (9, 4%), SSTI (9, 4%), respiratory (3, 1%), and UTI (1, 0.5%).
*Klebsiella pneumoniae*	17 (8%)	Primary BSI (7, 3%), SSTI (5, 2%), respiratory (3, 1%), UTI (1, 0.5%), and intra-abdominal infection (1, 0.5%).
*Proteus mirabilis*	9 (4%)	SSTI (5, 2%), UTI (1, 0.5%), primary BSI (1, 0.5%), and central nervous system (1, 0.5%).
*Proteus penneri*	3 (1%)	SSTI (3, 1%).
*Serratia marcescens*	8 (4%)	Primary BSI (4, 2%), respiratory (2, 1%), UTI (1, 0.5%), and SSTI (1, 0.5%).
*Morganella morganii*	7 (3%)	SSTI (5, 2%) and primary BSI (2, 1%).
*Enterobacter cloacae*	5 (2%)	Intra-abdominal infection (2, 1%), primary BSI (2, 1%), and respiratory (1, 0.5%).
*Haemophilus influenzae*	4 (2%)	Respiratory (4, 2%).
*Enterobacter aerogenes*	3 (1%)	SSTI (2, 1%) and primary BSI (1, 0.5%).
*Stenotrophomonas maltophilia*	3 (1%)	Respiratory (2, 1%) and primary BSI (1, 0.5%).
*Campylobacter jejuni*	2 (1%)	Intra-abdominal infection (2, 1%).
*Acinetobacter baumannii*	2 (1%)	Primary BSI (1, 0.5%) and respiratory (1, 0.5%).
*Acinetobacter lwoffii*	1 (0.4)	Respiratory (1, 0.5%).
*Providencia stuartii*	2 (1%)	Primary BSI (1, 0.5%) and SSTI (1, 0.5%).
*Achromobacter Xylosoxidans*	1 (0.5%)	Intra-abdominal infection (1, 0.5%).
*Enterobacter hormaechei*	1 (0.5%)	Primary BSI (1, 0.5%).
*Klebsiella oxytoca*	1 (0.5%)	SSTI (1, 0.5%).
*Pseudomonas stutzeri*	1 (0.5%)	Primary BSI (1, 0.5%).
*Sphingomonas paucimobilis*	1 (0.5%)	SSTI (1, 0.5%).
**Anaerobes**		
Acute *Clostridioides difficile* infection ^6^	4 (2%)	Intra-abdominal infection (4, 2%).
*Bacteroides fragilis*	3 (1%)	Intra-abdominal infection (1, 0.5%), SSTI (1,0.5%), and gynecological (pelvic) infection (1, 0.5%).
*Bacteroides uniformis*	1 (0.5%)	Intra-abdominal infection (1, 0.5%).
**Multidrug-resistant organisms (MDRO)**		
Ceftriaxone resistant Enterobacterales ^7^	36 (16%)	SSTI (14, 6%), UTI (7, 3%), primary BSI (6, 3%), respiratory (5, 2%), and intra-abdominal (4, 2%).
*Pseudomonas aeruginosa*	22 (10%)	Primary BSI (9, 4%), SSTI (9, 4%), respiratory (3, 1%), and UTI (1, 0.5%).
Methicillin-resistant *Staphyloccocus aureus* (MRSA)	11 (5%)	Primary BSI (9, 4%), respiratory (1, 0.5%), and intra-abdominal (1, 0.5%).
*Acinetobacter baumannii*	2 (1%)	Primary BSI (1, 0.5%) and respiratory (1, 0.5%).
Ampicillin-resistant *Enterococcus faecium*	1 (0.5%)	SSTI (1, 0.5%).
Ampicillin-resistant *Enterococcus raffinosus*	1 (0.5%)	SSTI (1, 0.5%).
Ceftriaxone-non-susceptible *Streptococcus mitis*	1 (0.5%)	Primary BSI (1, 0.5%).
**Extensively drug-resistant organisms (XDROs)**		
*Stenotrophomonas maltophilia*	3 (1%)	Respiratory (2, 1%) and primary BSI (1, 0.5%).
*Achromobacter Xylosoxidans*	1 (0.5%)	Intra-abdominal (1, 0.5%).
*Sphingomonas paucimobilis*	1 (0.5%)	SSTI (1, 0.5%).
Carbapenem-resistant *Pseudomonas aeruginosa*	2 (1%)	SSTI (2, 1%).
Carbapenem-resistant *Acinetobacter baumannii*	2 (1%)	Respiratory (1, 0.5%) and primary BSI (1, 0.5%).
Vancomycin-resistant enterococci (VRE)	2 (1%)	Intra-abdominal infection 2 (1%).

Note. BSI = bloodstream infection; UTI = urinary tract infection; SSTI = skin and/or soft tissue infection (e.g., cellulitis, septic arthritis, osteomyelitis, surgical site infection); MDRO = multidrug-resistant organism; XDRO = extensively drug-resistant organism. ^1^ The percent presented is among the 225 patients with microbiologically confirmed infection. ^2^ Coagulase-negative staphylococci, *Corynebacterium*, and *Lactobacillus* species could be determined as the cause of sepsis only in accordance with the skin contaminant criteria as issued by the Centers for Disease Control and Prevention (CDC) [6]. ^3^ Coagulase-negative staphylococci included *S. epidermidis* (*n* = 10), *S. Lugdunensus* (*n* = 1), and *S. waneri* (*n* = 1). ^4^ Streptococci bovis group includes *S. constellatus* (*n* = 2), *S. gallolyticus* (*n* = 3), and *S. infantarium* (*n* = 1). ^5^ Streptococci viridans group includes *S. dysgalactiae* (*n* = 1), *S. angiosum* (*n* = 1), and *S. mitis* (*n* = 1). ^6^ Acute *Clostridioides difficile* infection was determined in accordance with the Israeli Ministry of Health set of clinical and microbiological guidelines [7]. ^7^ Serves as a marker for extended-spectrum beta-lactamase (ESBL) production and/or *bla*_AmpC_ hyperproduction among Enterobacterales (with 99.2% sensitivity and 100% specificity) [8]. Includes *Escherichia coli* (*n* = 18), *Klebsiella pneumoniae* (*n* = 6), *Morganella morganii* (*n* = 3), *Proteus mirabilis* (*n* = 2), *Proteus penneri* (*n* = 2), and *Enterobacter aerogenes* (*n* = 2).

**Table 2 antibiotics-11-01255-t002:** Univariable analysis of septic hemodialysis adult patients with multidrug-resistant organism (MDRO) infections versus non-MDRO infections.

Parameter		MDRO * (*n* = 79) *n* (% **)	Non-MDRO (*n* = 430) *n* (% **)	MDRO vs. Non-MDRO	
				OR (CI-95%)	*p*-Value
**Demographics**					
**Age, years, mean ± SD**		**69 ± 14**	71 ± 13		0.28
Elderly (>65)		52 (66%)	306 (71%)	0.78 (0.5–1.3)	0.34
Male gender		54 (68%)	285 (66%)	1.1 (0.7–1.8)	0.72
**Dialysis parameters**					
Years on hemodialysis, years, median (range)		2 (0–30)	2 (0–31)		0.37
Chronic hemodialysis location	SMC outpatient clinic	43 (57%)	259 (61%)	0.86 (0.52–1.42)	0.56
	Out-of-hospital clinic	32 (43%)	166 (39%)		
Hemodialysis access	**A-V Fistula**	**17 (22%)**	**175 (41%)**	**0.4 (0.2–0.7)**	**0.002**
	A-V Graft	6 (8%)	24 (6%)	1.4 (0.6–3.6)	0.45
	**Perm-a-Cath**	**54 (70%)**	**230 (54%)**	**2 (1.2–3.4)**	**0.007**
Hemodialysis access site	**Arm**	**23 (30%)**	**198 (46%)**	**0.5 (0.3–0.8)**	**0.007**
	**Jugular**	**48 (62%)**	**205 (48%)**	**1.8 (1.1–2.9)**	**0.02**
	Femoral/Leg	7 (9%)	26 (6%)	1.5 (0.6–3.7)	0.33
	Abdominal wall	0	1 (0.2%)	N/A	>0.99
**Recent exposures to healthcare environments, settings, and procedures**					
Chronic resident of long-term care facility		11 (14%)	53 (12%)	1.2 (0.6–2.3)	0.69
**Recent (<3 months) hospitalization**		**48 (63%)**	**195 (46%)**	**2 (1.2–3.4)**	**0.005**
**Number of days from last hospitalization, median (IQ range)**		**65 (24–159)**	**102 (41–274)**		**0.012**
**Hospitalization in the past year**		**68 (91%)**	**341 (80%)**	**2.5 (1.1–5.5)**	**0.026**
**Skilled nursing home care *****		**6 (8%)**	**12 (3%)**	**2.9 (1.04–7.9)**	**0.034**
**Antibiotic exposure in the previous 3 months**		**45 (60%)**	**133 (31%)**	**3.3 (2–5.5)**	**<0.001**
**Invasive procedure in the preceding 6 months ******		**47 (62%)**	**181 (43%)**	**2.2 (1.3–3.6)**	**0.002**
**Permanent device *******		**58 (73%)**	**249 (58%)**	**2 (1.2–3.4)**	**0.01**
**MDRO* carrier in the past 2 years**		**45 (59%)**	**121 (28%)**	**3.7 (2.2–6.1)**	**<0.001**
**Background medical conditions**					
Dependent functional status [9]		46 (58%)	231 (54%)	1.2 (0.7–2)	0.46
Ischemic heart disease		49 (62%)	225 (52%)	1.5 (0.9–2.4)	0.112
Congestive heart failure		35 (44%)	200 (47%)	0.9 (0.6–1.5)	0.72
**Peripheral arterial disease**		**34 (43%)**	**101 (24%)**	**2.5 (1.5–4.1)**	**<0.001**
Diabetes mellitus		59 (75%)	305 (71%)	1.2 (0.7–2.1)	0.5
Chronic lung disease		19 (24%)	110 (26%)	0.9 (0.5–1.6)	0.77
Connective tissue disease		3 (4%)	29 (7%)	0.6 (0.2–1.8)	0.32
Liver disease		3 (4%)	33 (8%)	0.5 (0.1–1.6)	0.34
Past cerebrovascular attack (CVA or TIA)		13 (17%)	109 (25%)	0.6 (0.3–1.1)	0.09
Dementia		10 (13%)	61 (14%)	0.9 (0.4–1.8)	0.72
Immunosuppression		9 (11%)	50 (12%)	1 (0.5–2.1)	0.95
Active malignancy		5 (6%)	19 (4%)	1.5 (0.5–4)	0.46
**Chronic skin ulcer**		**36 (46%)**	**88 (21%)**	**3.3 (2–5.4)**	**<0.001**
Charlson’s score, median (IQ range) [10,11]	Weighted Comorbidity Index	6 (5–7)	6 (5–7)		0.51
	Combined Condition Score	8 (7–10)	9 (7–10)		0.47
	10-Year Survival	0 (0–0)	0 (0–0)		0.31
**Acute illness indices**					
Clinical syndrome	Dialysis access-related sepsis/primary bloodstream infection	34% (*n* = 27)	32% (136)	1.1 (0.7–1.9)	0.66
	**Respiratory infection**	**14% (*n* = 11)**	**37% (158)**	**0.3 (0.1–0.5)**	**0.0001**
	Urinary tract infection	10% (*n* = 8)	6% (24)	1.9 (0.8–4.4)	0.13
	**Skin and soft tissue infection**	**33% (*n* = 26)**	**17% (72)**	**2.4 (1.4–4.2)**	**0.0008**
	Abdominal and GI tract	9% (*n* = 7)	8% (34)	0.8 (0.4–2)	0.7
	Other	0	1% (6)	N/A	>0.99
Severe sepsis and septic shock		20 (25%)	102 (24%)	1.1 (0.6–1.9)	0.76
**Rapidly fatal McCabe condition [12]**		**22 (28%)**	**65 (15%)**	**2.2 (1.2–3.8)**	**0.006**
Fever (>38) or hypothermia		33 (42%)	169 (39%)	1.1 (0.7–1.8)	0.68
Tachycardia (>90 beats per minute)		31 (39%)	123 (29%)	1.6 (1–2.7)	0.059
Tachypnea (>20 min)		11 (14%)	85 (20%)	0.7 (0.3–1.3)	0.22
Leukocytosis (>12,000 cells/mm^3^) or leukopenia (<4000 cells/mm^3^)		34 (43%)	172 (40%)	1.1 (0.7–1.8)	0.61
**Serum albumin (g/dL)**		**31 ± 5**	**33 ± 5**		**<0.001**
Reduced consciousness at acute illness		18 (23%)	112 (26%)	0.8 (0.5–1.5)	0.54
Hospital division	**Internal medicine**	**54 (68%)**	**368 (86%)**	**0.4 (0.2–0.6)**	**0.0002**
	**Surgical ward**	**16 (20%)**	**40 (9%)**	**2.5 (1.3–4.7)**	**0.004**
	Obstetrics and gynecology	0	1 (0.2%)	N/A	>0.99
	**Intensive care unit**	**9 (11%)**	**21 (5%)**	**2.5 (1.1–5.7)**	**0.024**
Ventilated at time of culture		6 (8%)	27 (6%)	1.2 (0.5–3.1)	0.67
Pitt Score, median (IQ range) [13]		0 (0–1)	0 (0–1)		0.66
**Antimicrobials**					
Time to appropriate antimicrobial therapy (days), median (IQ range)		0 (0–1)	0 (0–0)		0.054
**Appropriate therapy in 48 h from culture**		**57 (73%)**	**127 (86%)**	**0.4 (0.2–0.9)**	**0.019**
**Misuse of broad-spectrum antibiotics for susceptible organism**		**6 (8%)**	**99 (68%)**	**0.4 (0.3–0.5)**	**<0.001**
**Outcomes**					
**Length of stay, days, median (IQ range)**		**10 (6–14)**	**7 (4–12)**		**<0.001**
In-hospital mortality		15 (19%)	49 (11%)	1.8 (1–3.4)	0.06
**Functional deterioration [14]**		**15 (23%)**	**41 (11%)**	**2.5 (1.3–4.9)**	**0.005**
Mortality in 90 days		25 (33%)	100 (24%)	1.5 (0.9–2.6)	0.12
Readmission in 3 months following discharge		35 (57%)	173 (48%)	1.5 (0.8–2.5)	0.18
*C. difficile* infection in 3 months following discharge		4 (7%)	7 (2%)	3.7 (1.1–13.2)	0.052

SD = standard deviation; MDRO= multidrug-resistant organism; SMC = Shamir Medical Center; GI= gastrointestinal; A-V= arterio-venous; N/A = not applicable; IQ= interquartile; CVA = cerebrovascular attack; TIA = transient ischemic attack; * MDROs include any one of the following: (1) *Staphylococcus aureus* resistant to oxacillin (i.e., MRSA), (2) ampicillin- and/or vancomycin-resistant enterococci (any enterococci), (3) penicillin or ceftriaxone-non-susceptible *Streptococcus pneumoniae*, (4) *A. baumannii* (regardless susceptibilities), (5) *P. aeruginosa* or *Achromobacter xylosoxidans* (regardless susceptibilities), (6) any Enterobacterales that is resistant to any 3rd- or 4th-generation cephalosporin (e.g., ceftriaxone, ceftazidime, cefotaxime, and cefepime), (7) any Enterobacterales with meropenem MIC > 1 (i.e., 2 or more), (8) metronidazole-non-susceptible anaerobic bacteria, (9) *Stenotrophomonas maltophilia*, and (10) *Campylobacter Jejuni* resistant to fluoroquinolones. ** Percentages are written as valid percent. *** Home intra-venous therapy or special wound care. **** Any type of invasive procedure, including any type of surgery (from minor to major, i.e., the whole spectrum), endoscopies, permanent central line insertions, any percutaneous procedure (e.g., coronary angiography), ascites paracentesis, and more. ***** Examples: permanent catheter (Perm-A-Cath), tracheotomies, tunneled central lines, silicon-based urinary catheters, orthopedic external fixators, implanted defibrillator, pacemaker, drains of any sort, and GI/urinary stoma.

## Data Availability

Not applicable.

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
