# Peer review of "The Epidemiology of Multidrug-Resistant Sepsis among Chronic Hemodialysis Patients"

_antibiotics, 2022, doi:10.3390/antibiotics11091255_

Round 1

Reviewer 1 Report

The manuscript by Shani Zilberman-Itskovich et al. describes interesting data on MDROs in the hemodialysis setting.

Many changes need to be made before possible acceptance for publication.

Type of article: original article and not communication I think.

Abstract: "avoided" is inappropriately underlined.

Global: prefer passive voice. Numbers below 12 should be written in capital letters. Italicize "i.e." or "e.g.".

Methods: did the authors type cephalospinase (Ampc). Phenotypic characterization may be insufficient.

Line 81: what do the authors mean by "(!)"?

Methods: How did the authors consider multiple inclusions for the same patients (as this could introduce a major bias) even if they are one month apart (I would recommend not including the same patient twice)? How did the authors consider correcting for multiple tests (as this could inflate the alpha risk)?

Figure 1: please change the legend, as it cannot be considered appropriate.

Line 181: please consider writing a sentence, not a period.

Line 217: how did the authors consider the constitution of the creation and validation cohorts? Please provide details.

Reviewer 2 Report

This is a paper based on a large number of cases and therefore can potentially provide various thoughts.

In my opinion, the most important element for evaluating the epidemiology of resistant organisms is the quantity and severity of previous infections (3-4 months) in which the same organism or from the same family, had emerged.

If the most significant group of patients of the paper is the one who had skin infections and/or peripheral arterial disease, it is probable that this is not the first treatment for these subjects.

Other typical causes of recurrent infection with development of resistant organisms in dialysis are native kidney infections (especially if polycystic), cholecystitis, non-functioning in situ graft, urinary retention infections and finally, the most frequent, those localized into permanent CVC; in the latter case it is necessary to know whether the CVC has been removed or not, because if it has colonized the development of resistant forms is the least that can happen.

The conclusions of the multivariate analysis (page 8) are too obvious to be of interest; the large number of items tested can be studied in deep.

In my opinion, if the authors want to get involved in more investigations, the paper should be improved and better characterized to draw more meaningful conclusions.

Round 2

Reviewer 1 Report

The comments have been appropriately adressed.

Reviewer 2 Report

In my opinion the large number of data analyzed could allow more interesting and targeted conclusions, these remain trivial and obvious. The work could have be improved becoming more trenchant.